# MODE CONNECTIVITY
# AND SPARSE NEURAL NETWORKS

## ABSTRACT

We introduce *instability analysis*, a framework for assessing whether the outcome of optimizing a neural network is robust to SGD noise. It entails training two copies of a network on different random data orders. If error does not increase along the linear path between the trained parameters, we say the network is *stable*.

Instability analysis reveals new properties of neural networks. For example, standard vision models are initially unstable but become stable early in training; from then on, the outcome of optimization is determined up to linear interpolation.

We leverage instability analysis to examine *iterative magnitude pruning* (IMP), the procedure underlying the lottery ticket hypothesis. On small vision tasks, IMP finds sparse *matching subnetworks* that can train in isolation from initialization to full accuracy, but it fails to do so in more challenging settings.

We find that IMP subnetworks are matching only when they are stable. In cases where IMP subnetworks are unstable at initialization, they become stable and matching early in training. We augment IMP to *rewind* subnetworks to their weights early in training, producing sparse subnetworks of large-scale networks, including Resnet-50 for ImageNet, that train to full accuracy.

## 1 INTRODUCTION

The lottery ticket hypothesis (Frankle & Carbin, 2019) conjectures that neural networks contain sparse subnetworks that are capable of training in isolation from initialization to full accuracy. The sole empirical evidence in support of the lottery ticket hypothesis is a series of experiments using a procedure called *iterative magnitude pruning* (IMP). IMP returns a subnetwork of the original, randomly initialized network by training the network to completion, pruning the lowest-magnitude weights (Han et al., 2015), and *resetting* each remaining weight to its original initialization. On small networks for MNIST and CIFAR-10, IMP subnetworks can match the accuracy of the full network (we refer to such subnetworks as *matching subnetworks*) at sparsity levels far beyond those at which randomly reinitialized or randomly pruned subnetworks can do the same.

The lottery ticket hypothesis offers a new perspective on the role of overparameterization and raises the tantalizing prospect that there may exist much smaller neural networks that are capable of replacing the larger models we typically train today. Unfortunately, in more challenging settings, there is no empirical evidence that the lottery ticket hypothesis holds. IMP subnetworks of VGG and Resnet-style networks on CIFAR-10 and ImageNet perform no better than other kinds of sparse networks (Liu et al., 2019; Gale et al., 2019).

In this paper, we describe a new framework called *instability analysis*, which measures whether the outcome of optimizing a network is robust to SGD noise (in which case we call it *stable*). Instability analysis offers a range of new insights into the behavior of unpruned networks. For example, the outcome of optimization becomes stable to SGD noise early in training (3% for Resnet-20 on CIFAR-10 and 20% on Resnet-50 for ImageNet). Moreover, it distinguishes known cases where IMP succeeds and fails to find a matching subnetwork; namely, IMP subnetworks are only matching when they are stable. It also allows us to identify new scenarios where sparse, matching subnetworks emerge early in training in more challenging settings, including Resnet-50 and Inception-v3 on ImageNet. In doing so, our results demonstrate that instability analysis is a valuable scientific tool for investigating the behavior of neural networks.

**Instability analysis.** Instability analysis is a technique to determine whether the outcome of optimization is robust to SGD noise. Figure 1 visualizes this process. We train two copies of the same network from initialization ($W_0$) on different data orders (which models different samples of SGD noise). We then linearly interpolate (dashed line) between the trained weights ($W_T^1$ and $W_T^2$) and examine the error along this path (blue curve). The *instability* of the network to SGD noise is the maximum increase in train or test error along this path (red line). We say a network is *stable* if error does not increase along the path, i.e., instability is 0. To examine instability at a later iteration $k$, we first train the network to iteration $k$ ($W_k$) and make two copies afterwards. Instability is a property of a network with respect to an optimization procedure; we focus on the standard procedure prescribed for the networks we examine.

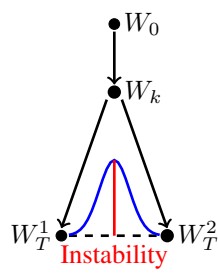

Figure 1: A diagram of instability analysis (text left).

Instability analysis assesses a linear form of *mode connectivity*, a phenomenon where the minima found by two networks are connected by a path of constant error. Draxler et al. (2018) and Garipov et al. (2018) show that the modes of standard vision networks trained from different initializations are connected by piece-wise linear paths of constant error or loss. Based on this work, we expect that our networks will be connected by such paths. However, the modes found by Draxler et al. and Garipov et al. are not connected by *linear* paths. The only extant example of linear mode connectivity is by Nagarajan & Kolter (2019), who train MLPs from the same initialization on disjoint subsets of MNIST and find that the resulting networks are connected by linear paths of constant test error; we explore linear mode connectivity from points throughout training, we do so at a larger scale, and we focus on different samples of SGD noise rather than disjoint samples of data.

**Results.** We begin by examining the instability of unpruned networks for MNIST, CIFAR-10, and ImageNet. All but the smallest MNIST network we study are unstable at initialization. However, by a point early in training (3% for Resnet-20 on CIFAR-10 and 20% for Resnet-50 on ImageNet), all networks become stable. In other words, from this point forward, the outcome of optimization is determined modulo linear interpolation. In fact, the entire trajectory of a stable network is so determined: when we train two copies of the network on different data orders, the states of the networks at each epoch are connected by linear paths over which test error does not increase.

In the lottery ticket context, we find that extremely sparse IMP subnetworks are matching only when they are stable, providing the first basis for understanding the mixed results in the literature. In doing so, we make a new connection between lottery ticket behavior and the optimization dynamics of neural networks. Inspired by our full network results, we modify IMP to *rewind* subnetwork weights to their values at iteration $k$ rather than *resetting* them to initialization. For values of $k$ that are early in training (in fact, earlier than the full networks), IMP subnetworks become stable in all cases we consider. Correspondingly, they also become matching. At these sparsity levels, randomly reinitialized and randomly pruned networks are neither stable nor matching. This connection between stability and accuracy suggests that linear mode connectivity is fundamental to sparse neural networks found by IMP and, thereby, to our current knowledge of the lottery ticket hypothesis.

**Contributions.** We make the following contributions:

- We introduce *instability analysis*, which measures the maximum increase in error along the linear path between minima found by training the same network on different data orders.

- On a range of image classification benchmarks including Resnet-50 on ImageNet, we observe that networks become stable to SGD noise early in training.

- We show that stable networks are stable throughout the training process.

- We use instability analysis to distinguish successes and failures of IMP (the core algorithm behind the lottery ticket hypothesis) as identified in previous work. Namely, extremely sparse IMP subnetworks are only matching when they are stable.

- We augment IMP with rewinding to study subnetworks initialized after iteration 0. We show that IMP subnetworks become stable and matching early in training if not at initialization.

- Using rewinding, we show how to find sparse, matching subnetworks in much larger settings than in previous work by setting the weights of IMP subnetworks to their values from early in training.

| Network | Dataset | Params | Train For | Batch | Accuracy | Opt | Rate | Schedule | Warmup | Density | Pruning |
|---------|---------|--------|-----------|-------|----------|-----|------|----------|--------|---------|---------|
| Lenet | MNIST | 266K | 24K Iters | 60 | $98.3 \pm 0.1\%$ | adam | 12e-4 | constant | 0 | 3.5% | Iterative |
| Resnet-20 (standard) | | | | | $91.7 \pm 0.1\%$ | mom. | 0.1 | | 0 | 16.8% | |
| Resnet-20 (low) | CIFAR-10 | 274K | 63K Iters | 128 | $88.8 \pm 0.1\%$ | mom. | 0.01 | 10x drop at 32K, 48K | 0 | 8.6% | Iterative |
| Resnet-20 (warmup) | | | | | $89.7 \pm 0.3\%$ | mom. | 0.03 | | 30K | 6.9% | |
| VGG-16 (standard) | | | | | $93.7 \pm 0.1\%$ | mom. | 0.1 | | 0 | 1.5% | |
| VGG-16 (low) | CIFAR-10 | 14.7M | 63K Iters | 128 | $91.7 \pm 0.1\%$ | mom. | 0.01 | 10x drop at 32K, 48K | 0 | 5.5% | Iterative |
| VGG-16 (warmup) | | | | | $93.4 \pm 0.1\%$ | mom. | 0.1 | | 30K | 1.5% | |
| Resnet-50 | ImageNet | 25.5M | 90 Eps | 1024 | $76.1 \pm 0.1\%$ | mom. | 0.4 | 10x drop at 30,60,80 | 5 Eps | 30% | Oneshot |
| Inception-v3 | ImageNet | 27.1M | 171 Eps | 1024 | $78.1 \pm 0.1\%$ | mom. | 0.03 | linear decay to 0.005 | 0 | 30% | Oneshot |

Table 1: Our networks and hyperparameters. Accuracies are the averages and standard deviations across three initializations. Hyperparameters for Resnet-20 (standard) are from He et al. (2016). Hyperparameters for VGG-16 (standard) are from Liu et al. (2019). Hyperparameters for *low*, *warmup*, and Lenet are adapted from Frankle & Carbin (2019). Hyperparameters for ImageNet networks are from Google's reference TPU code (Google, 2018).

## 2 PRELIMINARIES AND METHODOLOGY

**Instability analysis via linear mode connectivity.** Instability analysis evaluates whether the minima found when training two copies of a neural network on different samples of SGD noise (i.e., the random data order used during SGD) are linearly connected by a path over which error does not increase. The neural network in question could be randomly initialized ($W_0$ in Figure 1) or the result of $k$ training iterations ($W_k$). To perform instability analysis, we make two copies of the network and train them to completion with different random data orders ($W_T^1$ and $W_T^2$). We then linearly interpolate between the trained weights (dashed line) and compute the train or test error at each point (blue curve) to determine whether it increased (minima are not linearly connected) or did not increase (minima are linearly connected).

Formally, we capture training with SGD (or a variant) by a function $\mathcal{A}^{s \to t} : \mathbb{R}^D \times U \to \mathbb{R}^D$, which maps weights $W_s$ at iteration $s$ and SGD randomness $u \sim U$ to updated weights $W_t$ at iteration $t$ by training for $t - s$ steps (for $s, t \in \{1, .., T\}$ and $s < t$). Algorithm 1 describes our procedure:

---

**Algorithm 1** Stability analysis from iteration $k$.

1: Create a neural network with randomly initialized weights $W_0 \in \mathbb{R}^d$.
2: Train $W_0$ to $W_k$ under SGD noise $u \sim U$. That is, $W_k \leftarrow \mathcal{A}^{0 \to k}(W_0, u)$.
3: Train $W_k$ to $W_T^1$ under SGD noise $u_1 \sim U$. That is, $W_T^1 \leftarrow \mathcal{A}^{k \to T}(W_k, u_1)$.
4: Train $W_k$ to $W_T^2$ under SGD noise $u_2 \sim U$. That is, $W_T^2 \leftarrow \mathcal{A}^{k \to T}(W_k, u_2)$.
5: Evaluate $\mathcal{E}(\alpha W_T^1 + (1 - \alpha) W_T^2)$ for $\alpha \in [0, 1]$.

---

We describe the result of linear interpolation (step 5) with a quantity that we term *instability*. Let $\mathcal{E}(W)$ denote the train or test error of a network parameterized by $W$. Let $\bar{\mathcal{E}} = \text{mean}(\mathcal{E}(W_T^1), \mathcal{E}(W_T^2))$ be the average test error of $W_T^1$ and $W_T^2$. Let $\mathcal{E}_{\max} = \sup_{\alpha \in [0,1]} \mathcal{E}(\alpha W_T^1 + (1 - \alpha) W_T^2)$ be the highest test error when linearly interpolating between $W_T^1$ and $W_T^2$. The *instability* is $\mathcal{E}_{\max} - \bar{\mathcal{E}}$ (red line in Figure 1). When instability $\approx 0$, the minima are linearly connected and the network is *stable*. In practice, we average the instability from three initializations and three data orders per initialization (nine combinations in total). We use 30 evenly-spaced values of $\alpha \in [0, 1]$.

**Networks and datasets.** We study image classification networks on MNIST, CIFAR-10, and ImageNet as specified in Table 1. All hyperparameters listed are the standard values for these networks from reference implementations or prior work as cited in Table 1. The *warmup* and *low* variants of Resnet-20[1] and VGG-16 are adapted from hyperparameters chosen by Frankle & Carbin (2019).

---

[1] Frankle & Carbin (2019) mistakenly refer to Resnet-20 as "Resnet-18," which is a separate network.

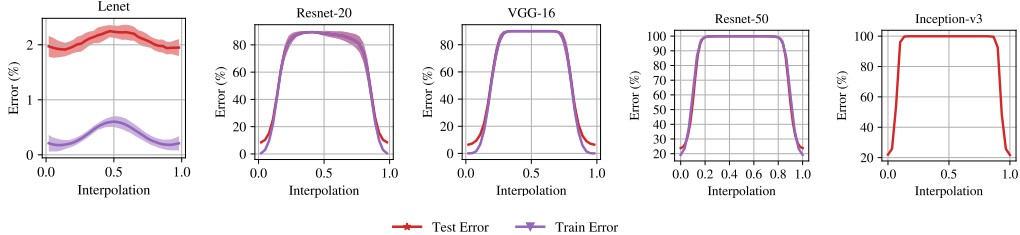

Figure 2: The train and test error when linearly interpolating between the minima found by randomly initializing a network and training it twice under different data orders. Each line is the mean and standard deviation across three initializations and three data orders (nine samples in total). The errors of the trained networks are at interpolation = 0 and 1. Inception train data is progress.

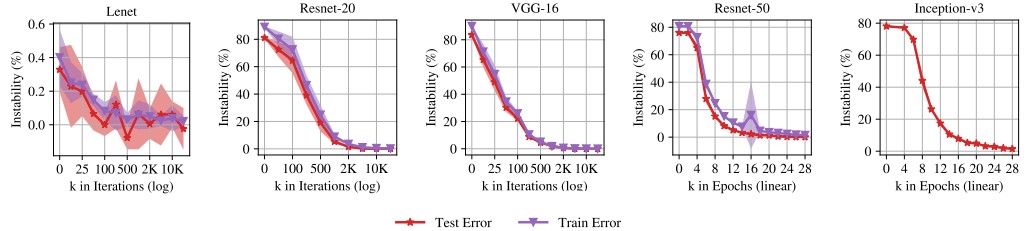

Figure 3: The instability when linearly interpolating between the minima found by networks that are trained on different data orders from step $k$. Each line is the mean and standard deviation across three initializations and three data orders (nine samples in total). Inception train data is progress.

## 3 NEURAL NETWORK INSTABILITY TO SGD NOISE

In this section, we study whether the outcome of optimization becomes robust to SGD noise after a certain amount of training. Concretely, we perform instability analysis (Algorithm 1) on the standard networks in Table 1 from many points during training to understand when, if ever, networks become stable to SGD noise. We find that, although only Lenet is stable at initialization, every network becomes stable early in training.

**Neural networks are unstable at initialization.** We begin by studying the instability of neural networks at initialization. We do so by training two copies of the same, randomly-initialized network under different samples of SGD noise (that is, Algorithm 1 with $k = 0$). Figure 2 shows the train error (purple) and test error (red) when linearly interpolating between the minima found by these copies. With the exception of Lenet on MNIST, none of the networks we study are stable at initialization. In fact, both training and test error rise to the point of random guessing when linearly interpolating between the minima found under different data orders. Lenet's error does rise slightly, but the increase is a small fraction of a percentage point. We conclude that, in general, larger-scale image classification networks are not stable at initialization.

**Stability improves early in training.** Although nearly all networks are unstable at initialization, they will inevitably become stable at some point. In the limit, they will be stable by the end of training, and it seems reasonable to expect that the final few steps of SGD are too insignificant to cause the network to enter linearly unconnected minima. In this experiment, we ask how early neural networks become stable. In other words, after what point in training is the outcome of optimization determined modulo linear interpolation regardless of the sample of SGD noise? To explore this behavior, we train a single copy of the network for $k$ iterations or epochs before making two copies that we train to completion on different data orders (Algorithm 1 with $k \geq 0$).

Figure 3 plots the *instability* of the networks for various values of $k$. We measure instability as the maximum error during interpolation (the peaks in Figure 2) minus the mean of the errors of the two networks (the endpoints in Figure 2). In all cases, instability decreases as $k$ increases, culminating in networks that are stable (i.e., instability $\approx 0$). The iteration at which stability emerges is surprisingly early. For example, it occurs from approximately iteration 2000 for Resnet-20 and VGG-16; in other words, after 3% of training, SGD noise cannot affect the final minimum modulo linear interpolation. Stability occurs later for Resnet-50: about epoch 18 (20% into training).

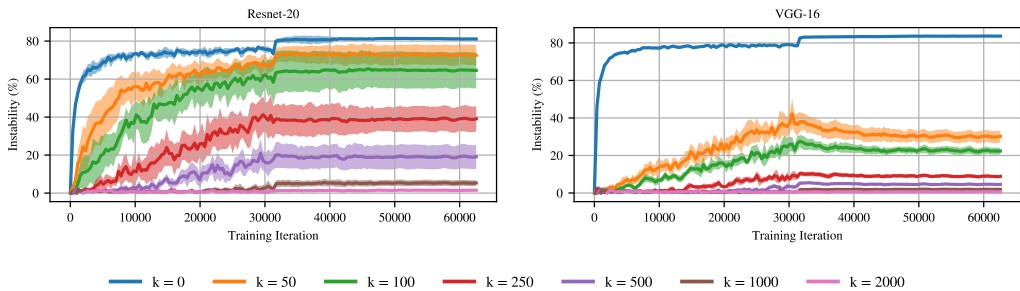

Figure 4: Test instability throughout training. At each epoch, linearly interpolate between the states of two networks trained with different SGD noise and compute instability. Each line involves training to iteration $k$ and then training two copies on different data orders after.

Instability is essentially identical when measured in terms of train or test error (although train instability is slightly higher than test instability for Resnet-50), indicating that the minimum becomes determined on both the train and test surfaces around the same time. Going forward, we present all results with respect to test error for simplicity.

**Stable networks are linearly connected throughout training.** Stable networks arrive at minima that are linearly connected, but do the trajectories they follow throughout training also have this property? In other words, when training two copies of the same network with different noise, is there a linear path over which test error does not increase connecting the states of the networks at each iteration? To study this behavior, we linearly interpolate between the networks at every epoch of training and compute the test error instability. That is, we compute instability *throughout* training.

Figure 4 plots instability throughout training for Resnet-20 and VGG-16 from different starting iterations $k$. For $k = 0$ (blue line), instability increases rapidly. In fact, it follows the same pattern as test error: as the test error of each network decreases, the maximum possible instability increases (since instability never exceeds random guessing). With larger values of $k$, instability increases more slowly throughout training. When $k$ is sufficiently large that the networks are stable at the end of training, they are stable at every epoch of training ($k = 2000$, pink line). In other words, after iteration 2000, the networks follow identical optimization trajectories modulo linear interpolation.

**Discussion.** Our observations implicitly divide training into two phases: an initial, unstable phase in which the final "linearly connected" minimum is undetermined on account of SGD noise and a subsequent, stable phase in which the final linearly connected minimum becomes determined. From this perspective, our observations contribute to a growing body of literature suggesting that training experiences a noisy initial phase and a less stochastic second phase. For example, the eigenspectrum of the Hessian settles into a bulk of small eigenvalues and a few large outlier eigenvalues after some amount of training (Gur-Ari et al., 2018), and networks trained with large batch sizes and high learning rates benefit from learning rate warmup during the first part of training (Goyal et al., 2017). One possible way to exploit our observations could be to explore changing aspects of the optimization process (e.g., learning rate schedule or optimizer) similar to Goyal et al. (2017) once the network enters the stable phase in order to improve the performance of training; instability analysis makes it possible to evaluate the consequences of doing so.

As a scientific tool, we also believe instability analysis provides a framework for studying topics related to the scale and distribution of SGD noise, e.g., the relationship between batch size, learning rate, and generalization (Keskar et al., 2017; Smith & Le, 2018; Smith et al., 2018) and the efficacy of alternative learning rate schedules (Smith, 2017; Smith & Topin, 2018; Li & Arora, 2019).

## 4    INSTABILITY AND SPARSITY

We have long known that it is possible to *prune* neural networks after training, often removing 90% of connections or more with no reduction in accuracy after small amount of additional training (e.g., LeCun et al., 1990; Reed, 1993; Han et al., 2015; Gale et al., 2019; He et al., 2018). However, sparse networks are more difficult to train from scratch. At the most extreme levels of sparsity

achievable by pruning, sparse networks trained in isolation generally reach lower test accuracy than dense networks (Han et al., 2015; Li et al., 2016; Liu et al., 2019; Frankle & Carbin, 2019).

However, there is a known class of networks that remains accurate at these sparsity levels: *winning lottery tickets*. On small vision networks, *iterative magnitude pruning* (IMP) retroactively finds sparse subnetworks that were capable of training in isolation to full accuracy (Frankle & Carbin, 2019); we refer to subnetworks with this capability as *matching subnetworks*. The existence of winning lottery tickets raises the possibility that we might be able to replace conventional, dense networks with sparser subnetworks, creating new opportunities to improve the performance of training. However, in more challenging settings, subnetworks found by IMP with $k = 0$ are not matching at particularly high sparsities and perform no better than other subnetworks (Liu et al., 2019; Gale et al., 2019). In these contexts, there is no evidence that the lottery ticket hypothesis holds.

Motivated by the possibility of training more efficient networks and a desire to explain the successes and failures of IMP, we study the relationship between instability and the accuracy of extremely sparse neural networks. Our central finding is that, although the accuracy of full networks in Section 3 seems unaffected by instability, the sparsest IMP subnetworks are matching only when they are stable. In other words, when SGD noise is sufficient to change the minimum that an IMP network finds (up to linear interpolation), test accuracy is lower. Randomly reinitialized and randomly pruned subnetworks are always both unstable and non-matching at all sparsity levels we consider.

## 4.1 METHODOLOGY

**Iterative magnitude pruning.** Iterative magnitude pruning (IMP) is a procedure to retroactively find a subnetwork of the state of the full network at iteration $k$ of training. As outlined in Algorithm 2, IMP trains a network to completion, prunes weights with the lowest magnitudes globally, and *rewinds* the remaining weights back to their values at iteration $k$. The result is a subnetwork $(W_k, m)$ where $W_k \in \mathbb{R}^d$ is the state of the full network at iteration $k$ and $m \in \{0, 1\}^d$ is a fixed binary vector that, when multiplied element-wise with $W_k$, produces the pruned network $m \odot W_k$. We can either run IMP iteratively (training, pruning 20% of weights (Han et al., 2015; Frankle & Carbin, 2019), rewinding, and repeating until we reach a target sparsity) or in one-shot (pruning to the target sparsity in a single step). We use one-shot pruning on ImageNet networks for efficiency and iterative pruning in all other cases (Table 1). Frankle & Carbin (2019) only study rewinding to iteration 0; one of our contributions is to generalize IMP to any rewinding iteration $k$. When training a subnetwork from iteration $k$, we also rewind the learning rate schedule to its state at iteration $k$.

---

**Algorithm 2** Iterative Magnitude Pruning (IMP) with rewinding to iteration $k$ and $N$ iterations.

1: Create a neural network with randomly initialized weights $W_0 \in \mathbb{R}^d$ and initial pruning mask $m = 1^d$.
2: Train $W_0$ to $W_k$ under SGD noise $u \sim U$. That is, $W_k \leftarrow \mathcal{A}^{0 \rightarrow k}(W_0, u)$.
3: **for** $n \in \{1, \ldots, N\}$ **do**
4:     Train $m \odot W_k$ to $m \odot W_T$ under SGD noise $u' \sim U$. That is, $W_T \leftarrow \mathcal{A}^{k \rightarrow T}(m \odot W_k, u')$.
5:     Prune the remaining entries with the lowest magnitudes from $W_T$. Let $m[i] = 0$ if $W_T[i]$ is pruned.
6: Return $m, W_k$

---

**Sparsity levels.** Although we are interested in the behavior of networks at all sparsities, computational limits force us to focus on a specific sparsity level.[2] In light of these restrictions, we focus on the highest sparsities for which IMP returns a matching network at any rewinding iteration $k$. The densities we examine are in Table 1, and Appendix A explains these choices. Doing so provides the best contrast between sparse networks that are matching and (1) the full, overparameterized neural networks and (2) other classes of sparse networks.

**Experimental approach.** We study the relationship between stability and accuracy in extremely sparse subnetworks uncovered by IMP. IMP produces particularly sparse matching subnetworks and is the algorithm behind current lottery ticket results, so it merits close examination for both better scientific understanding and potential practical lessons for training sparse networks. As a basis

---

[2]IMP entails training a network at least a dozen times to reach high levels of sparsity, and instability analysis requires training each of these networks a further nine times (three data orders and three kinds of sparsity) for many rewinding iterations. For rigor, we replicate each experiment three times with different initializations.

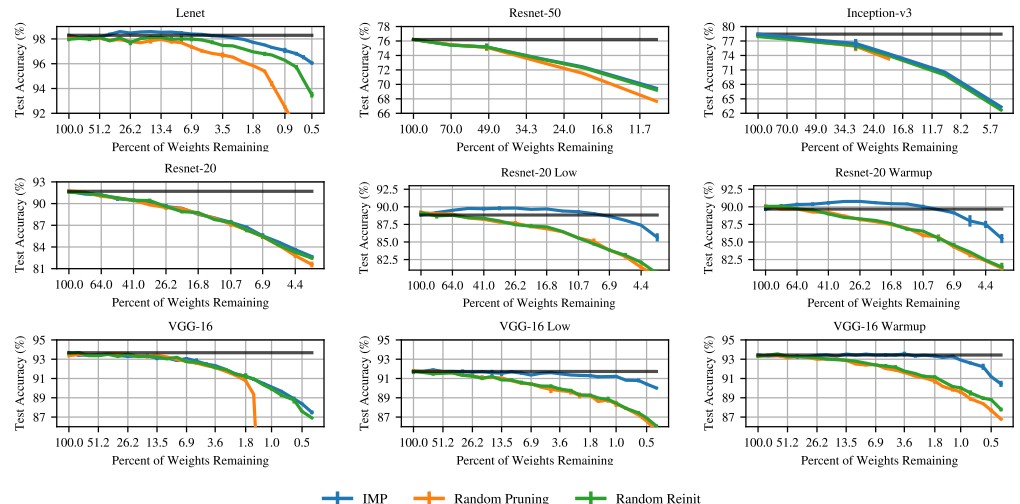

Figure 5: Test accuracy of IMP subnetworks, randomly pruned subnetworks, and randomly reinitialized IMP subnetworks at all levels of sparsity. The black line is the accuracy of the unpruned network. Each line is the mean and standard deviation across three initializations.

for comparison, we also examine two kinds of subnetworks that are not matching at the sparsities we consider: (1) IMP subnetworks that are randomly reinitialized and (2) subnetworks found by randomly pruning weights rather than pruning those with the lowest magnitudes. We exploit the fact that not all IMP subnetworks are matching: we contrast settings where IMP succeeds and fails to further understand the conditions under which IMP subnetworks are matching.

## 4.2 Experiments

**IMP subnetworks are matching at initialization only when stable.** We begin by studying sparse subnetworks trained in isolation from initialization ($k = 0$). As noted previously, not all IMP subnetworks are matching at the sparsity levels we consider for $k = 0$. Figure 5 shows the accuracy of the IMP subnetworks (blue) across all levels of sparsity for each of the hyperparameters in Table 1 (alongside randomly pruned subnetworks in orange and randomly reinitialized subnetworks in green for comparison). On Lenet, IMP subnetworks are matching at sparsities well beyond those at which other subnetworks are matching. The same is true for variants of Resnet-20 and VGG-16 with lower learning rates or learning rate warmup, changes proposed by Frankle & Carbin (2019) specifically to make it possible for IMP to find matching subnetworks. In contrast, IMP subnetworks of Resnet-50, Inception-v3, and standard configurations of Resnet-20 and VGG-16 perform similarly to randomly reinitialized and randomly pruned subnetworks.

In Figure 6, we analyze the instability of these subnetworks. At the sparsity levels we consider, IMP subnetworks are matching only when they are stable. The IMP subnetworks of Lenet, Resnet-20 (low, warmup), and VGG-16 (low, warmup) are stable and matching, while no other IMP subnetworks have either property. The low and warmup experiments are notable because these hyperparameters were selected by Frankle & Carbin (2019) to make it possible for IMP to find matching subnetworks without awareness that they also improve stability. This inadvertent causal experiment adds further evidence of a connection between instability and accuracy in IMP subnetworks.

With the exception of Lenet, no randomly reinitialized or randomly pruned subnetworks are stable or matching at these levels of sparsity. On Lenet, these subnetworks are not matching but test error only rises slightly when interpolating. For all other networks we consider, the error of these subnetworks approaches or reaches that of random guessing when interpolating.

**IMP subnetworks become stable and matching early in training.** In the previous experiment, we saw that the IMP subnetworks are matching only when they are stable to SGD noise. In Section 3, we observed that unpruned networks become stable to SGD noise only after a certain amount of training. In this experiment, we combine these observations: we study whether IMP subnetworks become stable during training and, if so, whether improved accuracy follows. To do so, we examine

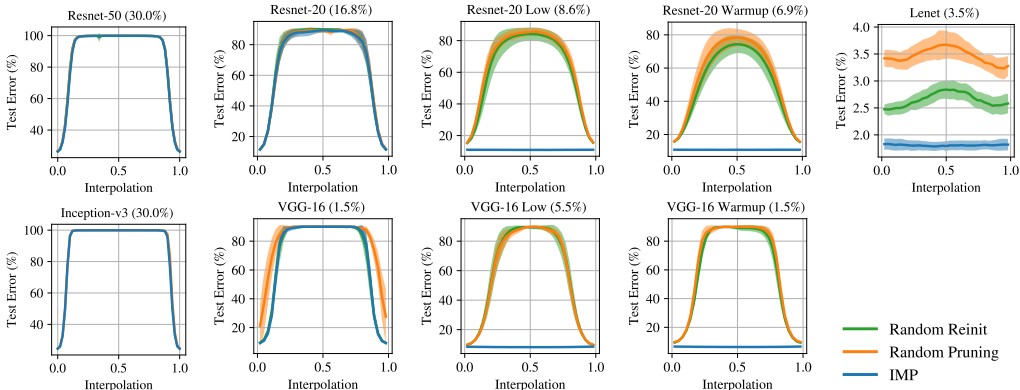

Figure 6: The test error when linearly interpolating between the minima found by training sparse subnetworks twice from initialization under different data orders. Each line is the mean and standard deviation across three initializations and three data orders (nine samples in total). The test errors of the trained networks are at 0 and 1. Percents are percents of weights remaining.

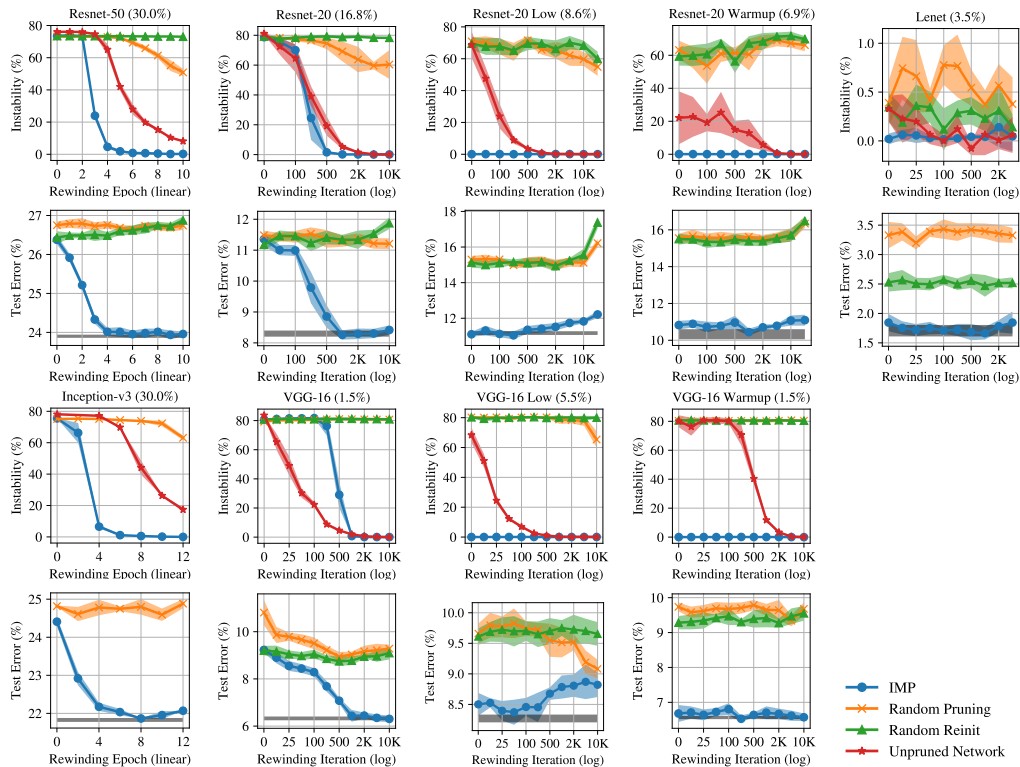

Figure 7: The instability of subnetworks that are created from the state of the full network at iteration $k$ and trained on different data orders from there. Each line is the mean and standard deviation across three initializations and three data orders (nine samples in total). The gray lines are the accuracies of the full networks to one standard deviation. Percents are percents of weights remaining.

the sparse subnetworks that result from training unpruned networks for $k$ steps and subsequently applying pruning masks (and possibly reinitializing). We find these masks by running IMP with *rewinding*: we train the full network to completion, prune, and *rewind* each remaining weight to its value at step $k$. We then run standard instability analysis on these sparse networks from iteration $k$.

The blue dots in Figure 7 show the instability (rows 1 and 3) and test accuracy (rows 2 and 4) when rewinding IMP subnetworks to various points early in training. Those subnetworks that are unstable when rewound to iteration 0 (Resnet-20, VGG-16, Resnet-50, Inception-v3) become stable when rewound to points slightly later in training. IMP subnetworks of Resnet-20, VGG-16, and Resnet-50

become stable at about iteration 500 (0.8% into training), iteration 1000 (1.6%), and epoch 4 (4.4%). Stability and accuracy of these sparse IMP subnetworks continue to correlate. Test error decreases alongside instability, with IMP subnetworks reaching the performance of the unpruned networks (gray lines) approximately when they become stable. IMP subnetworks that were matching and stable at iteration 0 generally remain so at other rewinding points, although Resnet-20 low and VGG-16 low experience increased test error at the latest rewinding points we consider.

IMP subnetworks become stable at least as early as the unpruned networks (red) and much earlier for Resnet-50 (epoch 4 vs. 18). In contrast, randomly pruned subnetworks (orange) and randomly reinitialized IMP subnetworks (green) are unstable and non-matching at every rewinding iteration (with Lenet again the sole exception). We believe these subnetworks will eventually become stable later on; in some cases, instability of randomly pruned subnetworks decreases at the latest rewinding points we consider. This behavior suggests a potential broader link between subnetwork stability and accuracy: IMP subnetworks are matching and maintain or improve upon the stability behavior of the full networks, while other subnetworks are less accurate and become stable later if at all.

## 4.3 DISCUSSION

**The "lottery ticket hypothesis."** The lottery ticket hypothesis (Frankle & Carbin, 2019) conjectures that any "randomly initialized, dense neural network contains a subnetwork that—when trained in isolation—matches the accuracy of the original network." The authors support this hypothesis by using IMP to find matching subnetworks at initialization in small vision networks. However, follow-up studies show (Liu et al., 2019; Gale et al., 2019) and we confirm that IMP does not find matching subnetworks in more challenging settings. We use instability analysis to distinguish the successes and failures of IMP as identified in previous work. In doing so, we make a new connection between the lottery ticket hypothesis and the optimization dynamics of neural networks.

Moreover, by augmenting IMP with rewinding, we show how to find sparse, matching subnetworks in much larger settings than in previous work, albeit with subnetworks from *early* in training rather than at initialization. Our technique has already been adopted to create trainable subnetworks that transfer to new settings (Morcos et al., 2019), as a pruning method in its own right (Anonymous, 2020a), and to further study the lottery ticket hypothesis (Anonymous, 2020e;c;g;f;b).

**Pruning.** On larger-scale networks and tasks, we find that IMP subnetworks at extreme sparsities only become stable and matching after the full network has been trained for a small number of iterations or epochs. Recent methods have explored pruning neural networks at initialization (Lee et al., 2019; Anonymous, 2020d), but our results suggest that the best time to prune may be slightly later in training. By that same token, most modern pruning methods only begin to sparsify networks late in training or after training (Han et al., 2015; Gale et al., 2019; He et al., 2018). In these cases, the fact that there are matching subnetworks early in training suggests that there is potentially a substantial unexploited opportunity to prune neural networks much earlier than current methods.

**SGD noise and overparameterization.** While dense neural networks train to full accuracy regardless of their stability, sparse networks in our experiments are only matching when they are stable. Although our results speak only to specific kinds of sparse networks (IMP subnetworks and our randomly reinitialized and randomly pruned baselines) at particularly extreme sparsity levels, they suggest a possible broader relationship between instability and accuracy of sparse networks. It is possible that sparse networks, which have fewer parameters than their dense counterparts, are less robust to instability during the early part of training.

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

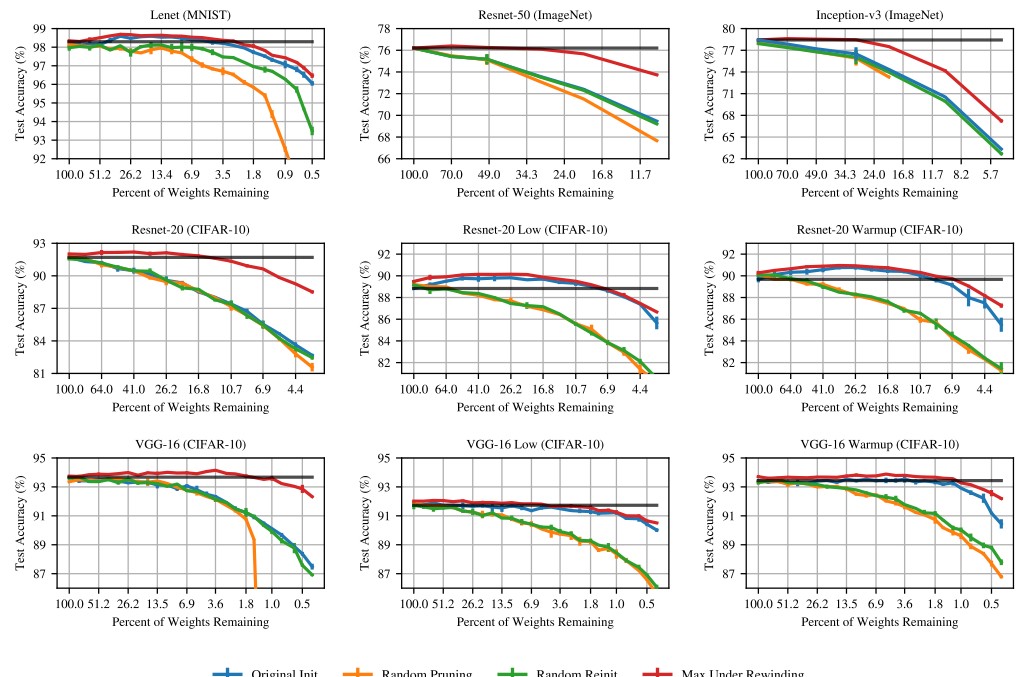

Figure 8: At all sparsity levels, the maximum test accuracy achieved by IMP subnetworks for any rewinding iteration (red). Also includes the test accuracy of IMP subnetworks with $k = 0$, randomly pruned subnetworks, and randomly reinitialized IMP subnetworks with $k = 0$ at all levels of sparsity. The black line is the accuracy of the unpruned network. Each line is the mean and standard deviation across three initializations.

## A    SELECTING EXTREME SPARSITY LEVELS FOR IMP SUBNETWORKS

In this appendix, we describe how we select the sparsity level that we examine for each IMP subnetwork. For each network and hyperparameter configuration, our goal is to study the most extreme sparsity level at which matching subnetworks are known to exist early in training. To do so, we use IMP to generate subnetworks at many different sparsities for many different rewinding iterations (specifically, all of the rewinding iterations Figure 7). We then select the most extreme sparsity level at which *any* rewinding iteration produces a matching subnetwork.

Figure 8 plots the maximum accuracy found by any rewinding iteration in red. The black line is the accuracy of the unpruned network to one standard deviation. For each network, we select the most extreme sparsity for which the red and black lines overlap. As a basis for comparison, Figure 8 also includes all of the other lines from Figure 5: the result of performing IMP with $k = 0$ (blue line), random pruning (orange line), and random reinitialization of the IMP subnetworks with $k = 0$ (green line).

Note that, for computational reasons, Resnet-50 and Inception-v3 are pruned using *one-shot pruning*, meaning the networks are pruned to the target sparsity all at once. All other networks are pruned using *iterative pruning*, meaning the networks are pruned by 20% after each iteration of IMP until they reach the target sparsity.

## B    INTERPOLATION DATA FOR UNPRUNED NETWORKS

In this appendix, we present the interpolation data for the instability analysis on the unpruned networks in Section 3.

## B.1 TEST ERROR

These graphs plot the test error when linearly interpolating for select values of $k$ for the networks in Figure 3.

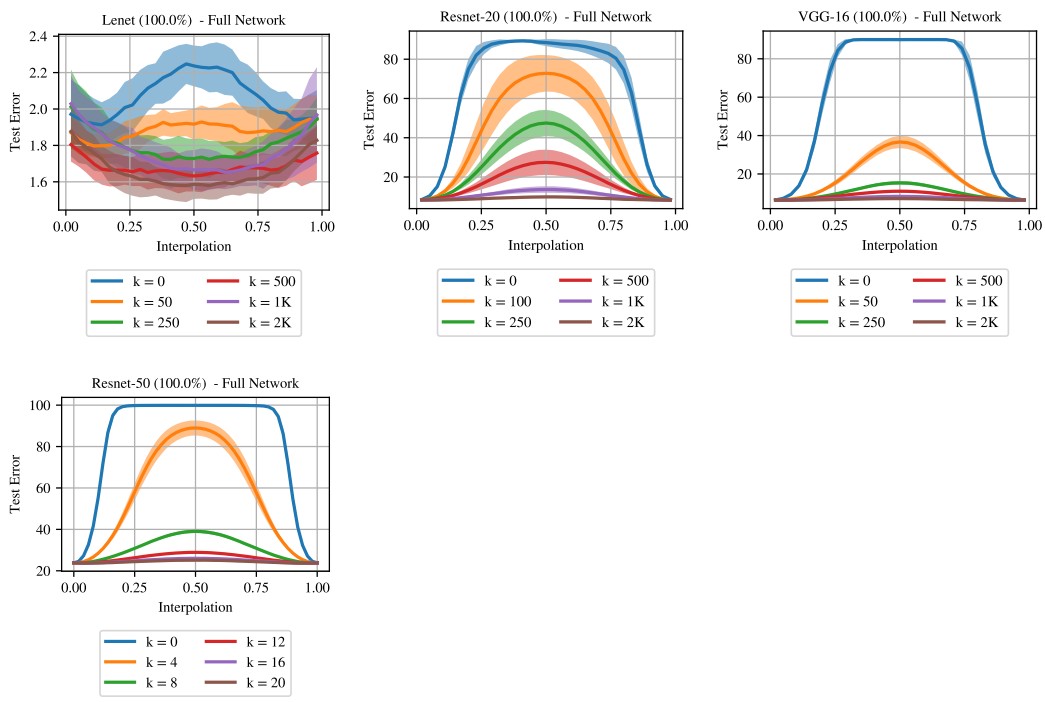

## B.2 TRAIN ERROR

These graphs plot the train error when linearly interpolating for select values of $k$ for the networks in Figure 3.

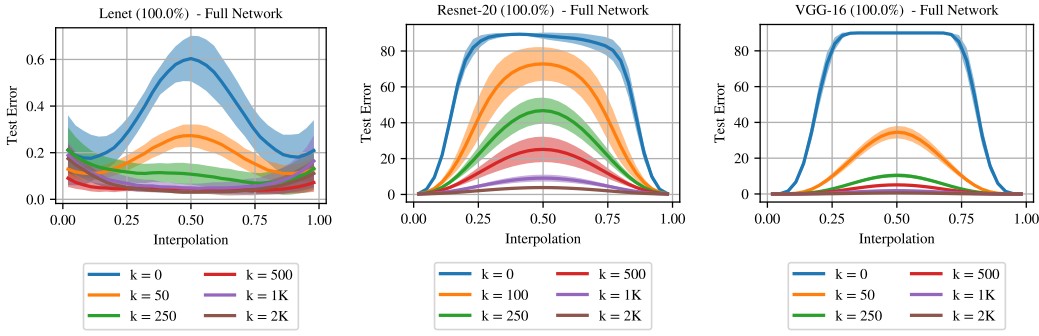

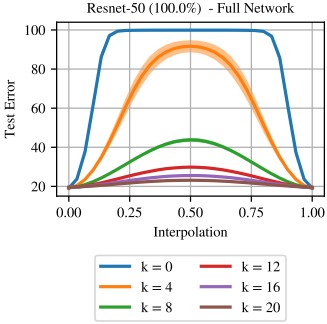

# C    INTERPOLATION DATA FOR SPARSE NETWORKS

In this appendix, we present the interpolation data for the instability analysis on the sparse networks in Section 4.

## C.1    TEST ERROR OF IMP SUBNETWORKS

These graphs plot the test error when linearly interpolating for select values of $k$ for the IMP subnetworks in Figure 7. Percents in all figures are *densities*—the percent of weights remaining after pruning.

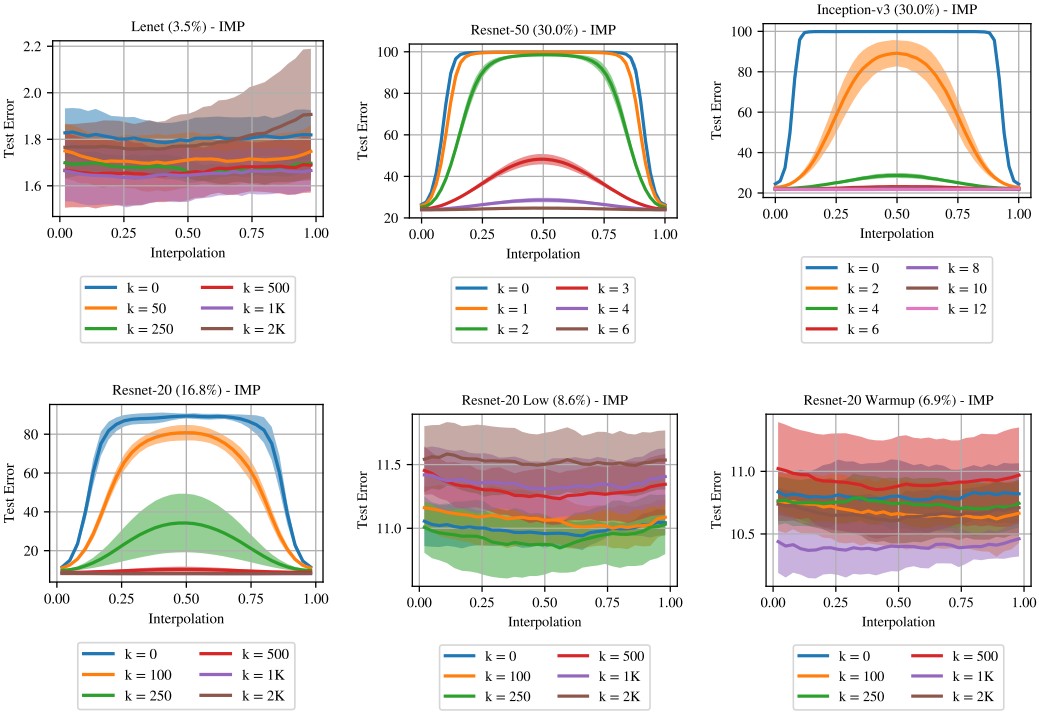

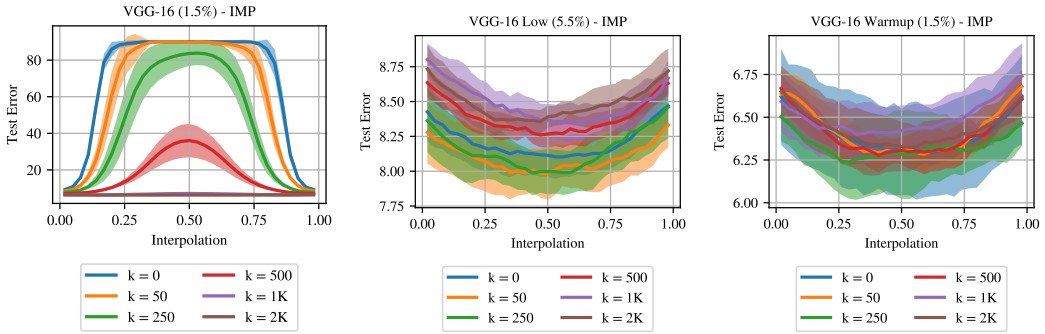

## C.2 TEST ERROR OF RANDOMLY PRUNED SUBNETWORKS

These graphs plot the test error when linearly interpolating for select values of $k$ for the randomly pruned subnetworks in Figure 7. Percents in all figures are *densities*—the percent of weights remaining after pruning.

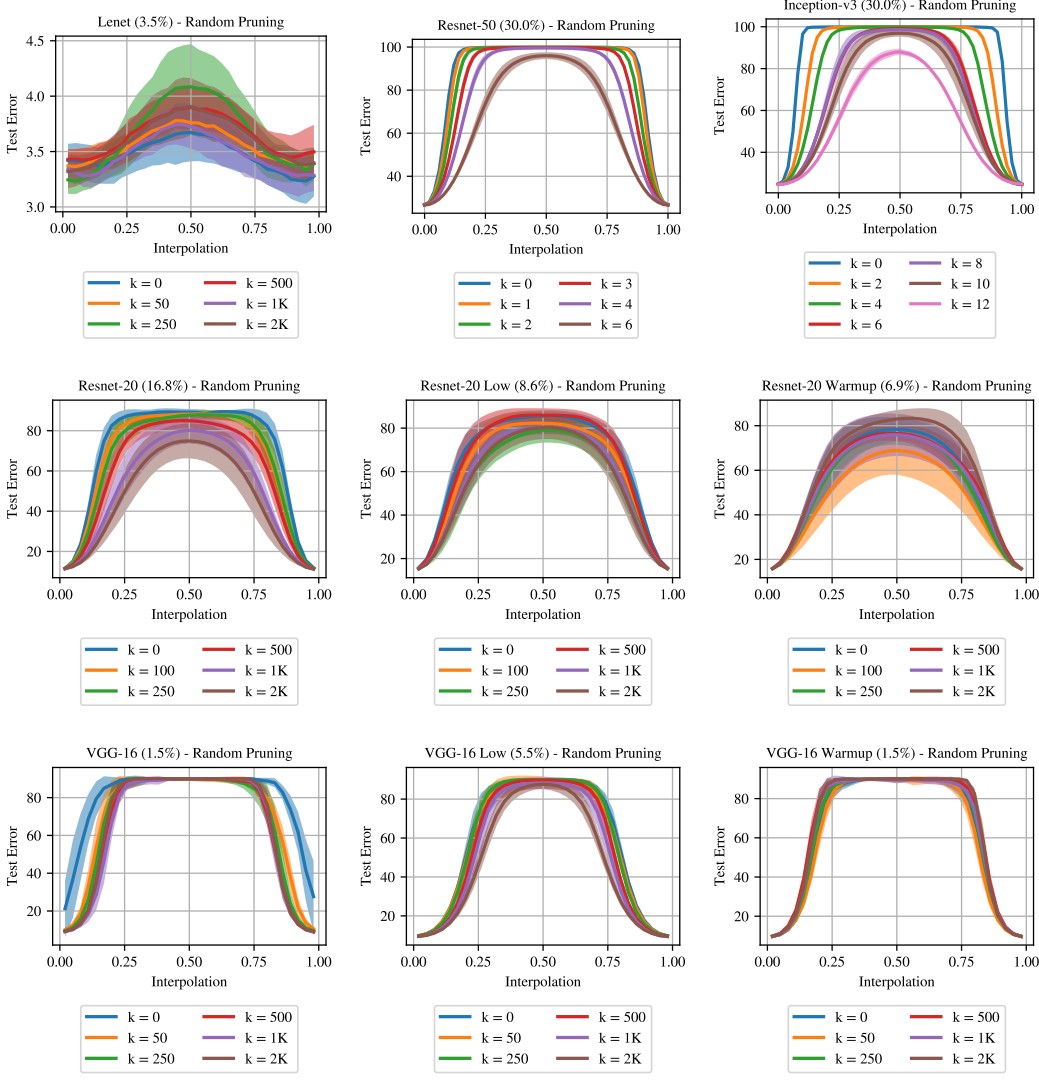

### C.3    Test Error of Randomly Reinitialized IMP Subnetworks

These graphs plot the test error when linearly interpolating for select values of $k$ for the randomly reinitialized IMP subnetworks in Figure 7. Percents in all figures are *densities*—the percent of weights remaining after pruning.

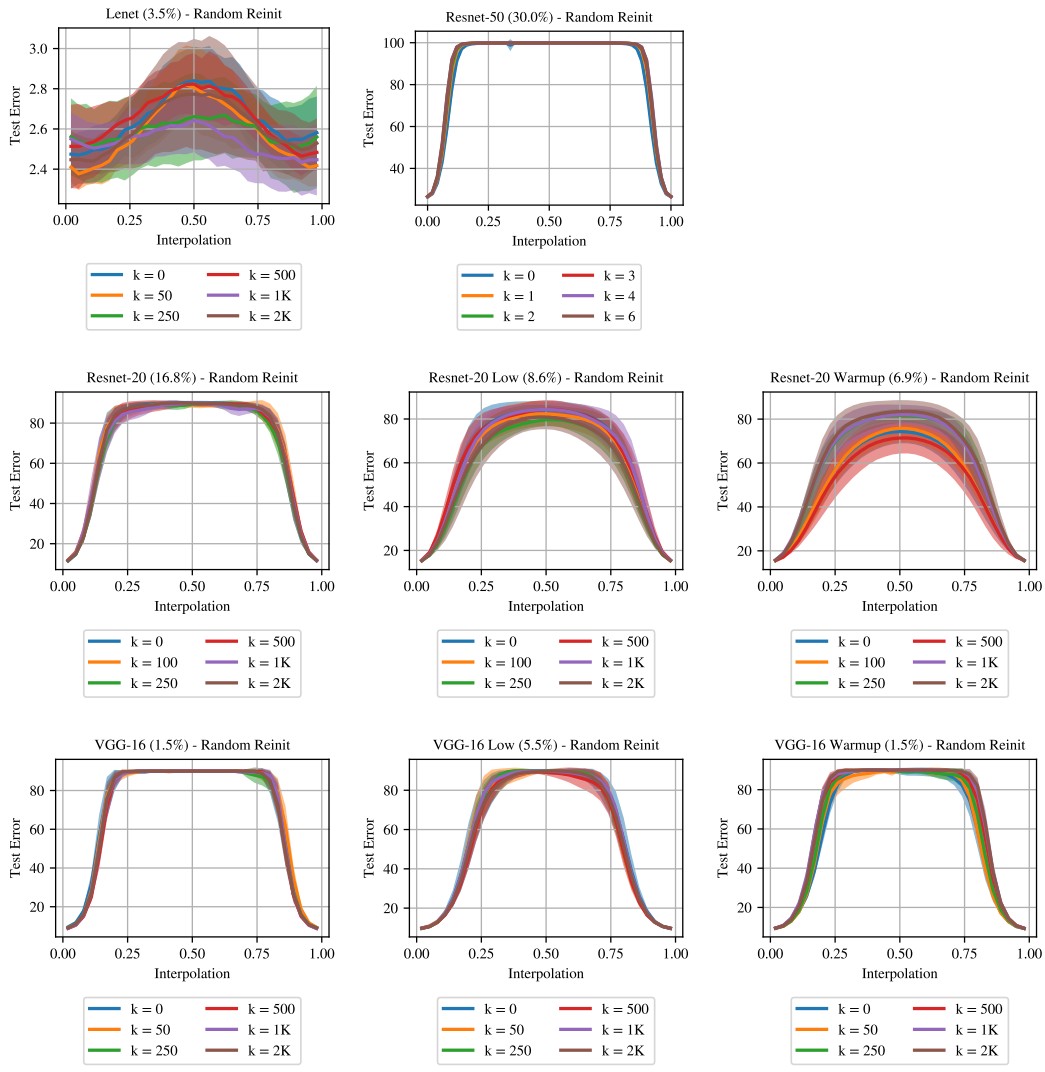

## D    $L_2$ Distances for Unpruned Networks

In this appendix, we present the $L_2$ distances between pairs of full networks trained on different data orders from iteration $k$, the experiment in Section 3. This data parallels Figure 3. We do not yet have $L_2$ distance data for the ImageNet networks, although we plan to add it to the next version of the paper.

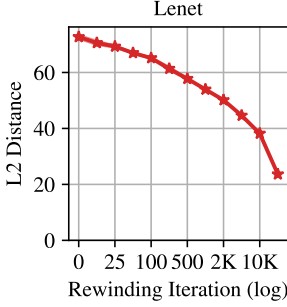 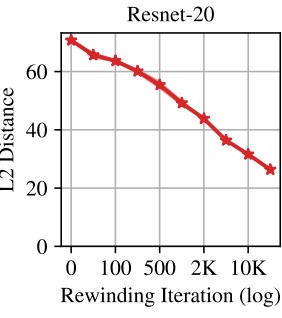 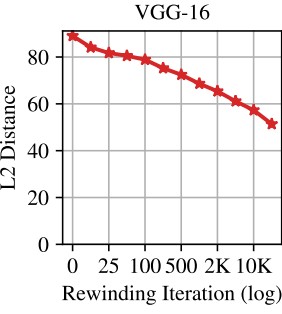

The $L_2$ distance between the networks decreases linearly as $k$ increases. Interestingly, we observe no clear relationship between the $L_2$ distance and the network instability. For example, there does not appear to be a critical $L_2$ distance threshold that is crossed when the networks become stable. This is in contrast to our observations in Appendix E, where $L_2$ distance between IMP networks correlates with instability, dropping to a lower value when the subnetworks become stable.

# E  $L_2$ DISTANCES FOR SPARSE NETWORKS

In this appendix, we present the $L_2$ distances between pairs of sparse networks trained on different data orders from iteration $k$, the experiment in Section 4. This data parallels Figure 7. We do not yet have $L_2$ distance data for the ImageNet networks, although we plan to add it to the next version of the paper.

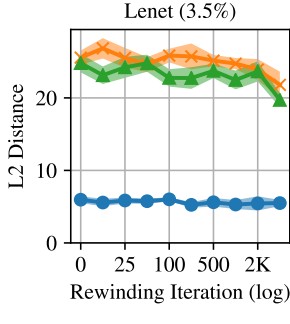

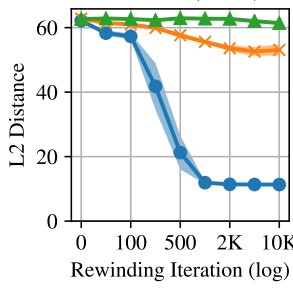 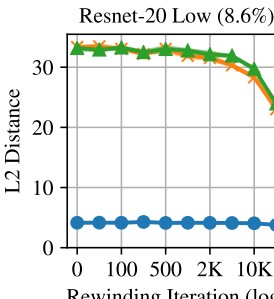 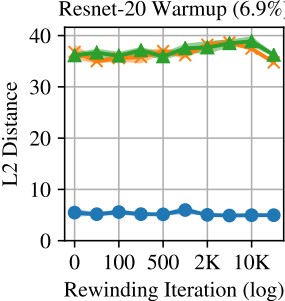

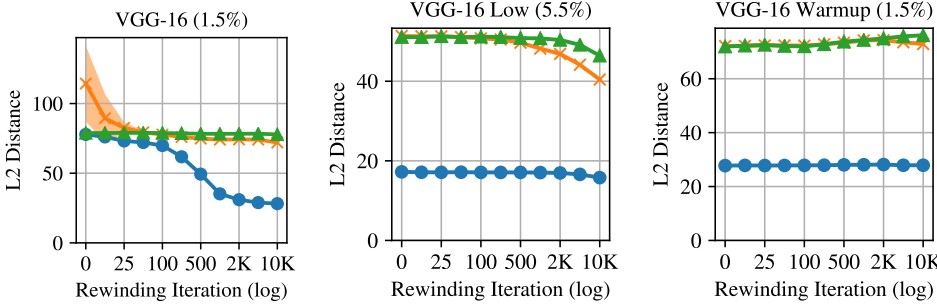

The $L_2$ distance between IMP subnetworks follows the same pattern as instability. When the network is unstable, the $L_2$ distance plateaus at a higher level, the same level as randomly reinitialized and randomly pruned networks. As instability decreases, $L_2$ distance also decreases. When the subnetwork becomes stable, $L_2$ distance plateaus at a lower level than the randomly reinitialized and randomly pruned networks. Importantly, this lower level is still non-zero. These results contrast with thise in Appendix D, where we do not observe a relationship between instability and $L_2$ distance between the full networks.

