# OpenReview forum: "Mode Connectivity and Sparse Neural Networks"
_ICLR.cc/2020/Conference — Reject_

### Official Review · AnonReviewer3 · 2019-10-11
**Official Blind Review #3**

**Rating:** 3

**Review:**

This paper works on empirically demonstrating the connection between model connectivity and the lottery ticket hypothesis, which are individually explored in the literature. Here the model connectivity refers to the fact that SGD produces different solutions (from the randomness, such as data ordering) that are connected through model parameter transition paths of approximately equal loss/accuracy. The lottery ticket hypothesis tells that there exist sparse subnetworks of the corresponding full dense network which can attain as strong loss / accuracy as the full dense network.

As the primary contribution, the authors demonstrated that the following two observations often emerge together: 1) A sparse subnetwork can match the performance of the corresponding full dense network;  2) Running SGD on this sparse subnetwork produces solutions which are connected via a linear model parameter transition path of similar loss/accuracy; this is observed across both small tasks (using CIFAR10) and ImageNet-level tasks. Another contribution I can see besides the primary one is that the lottery ticket hypothesis still holds on large tasks, which is against the conventional wisdom demonstrated in recent papers (e.g. Gale et al., 2019); the authors show that it needs to rewind to weights after a short period of training instead of rewinding to the initialized weight in Iterative Magnitude Pruning to produce the "lottery ticket" in large tasks (such as CNN for ImageNet).

I think the primary contribution on the connection between model connectivity and lottery ticket hypothesis is an interesting observation, but the content are poorly presented for me fully appreciate the importance and practical implications of this work. Thus I give weak reject. The major concerns and questions are as the following:

1. From the paper, I don't understand why the connection between model connectivity and lottery ticket hypothesis is an important one to reveal. Is it important because it implies some practical approaches / heuristics to figure out performant sparse subnetworks? Is it intrinsically interesting because it validates some hypothesis in the training dynamics of SGD? These are not clear to me.

2. I think the current presentation of the content is only limited to the empirical demonstration. And I can not extract useful intuitions/messages from the demonstration here on why this happens. These message should provide intuitions on why this connection exists. E.g. These message can be extracted from some SGD on some simple (toy) non-convex models with multiple local minimum regions.

Minor comments for improving the paper:

1. At the end of line 1 in algorithm one, it is not clear what 1^|W0| means.

2. The terms in figure legend needs to be properly defined to enable clear reading. Currently words such as "reset" is not mentioned in the text but appears in the legend of figure 4 and etc.











**Experience Assessment:**

I have read many papers in this area.

**Review Assessment: Checking Correctness Of Derivations And Theory:**

N/A

**Review Assessment: Checking Correctness Of Experiments:**

I assessed the sensibility of the experiments.

**Review Assessment: Thoroughness In Paper Reading:**

I read the paper at least twice and used my best judgement in assessing the paper.

---

> ### Author Response · Authors · 2019-11-14
> **Author Response to Reviewer 3**
>
>
> NOTE: We have posted an updated version of the paper that has been substantially restructured and rewritten to address your concerns. We highly recommend looking over the new paper.
>
> We have summarized these changes in a general response (posted as a top-level comment). We ask that you read our general response before returning to this point-by-point response. We address many of your concerns there.
>
> --------------------
>
> > The content are poorly presented for me fully appreciate the importance and practical implications of this work
>
> We apologize that our original presentation did not clearly articulate the importance and practical implications of our work. We have taken your feedback to heart, and we have substantially restructured and rewritten the paper to ensure that these aspects are clear. We have summarized our clarified framing in the top-level comment.
>
> We specifically address practical implications in the PRACTICAL IMPLICATIONS section of the top-level comment and in the discussion sections of our revised paper.
>
> > I don't understand why the connection between mode connectivity and lottery ticket hypothesis is an important one to reveal.
>
> In the top-level comment, we present our clarified framing for the paper designed to emphasize why both linear mode connectivity on full networks and its connection to the lottery ticket hypothesis are important. We discuss concrete practical implications of our observations in the PRACTICAL IMPLICATIONS section of the top-level comment.
>
> > I can not extract useful intuitions/messages from the demonstration here on why this happens.
>
> At the moment, the phenomena we observe are entirely empirical. We do not yet have a theoretical model to describe this behavior, although we are exploring various connections (e.g., it is consistent with the so-called neural tangent kernel regime where very wide neural networks behave like linear models). However, we contend that our experiments are sufficiently rigorous to convincingly establish the existence of these phenomena. We believe that recording these phenomena rigorously is a significant contribution that will inspire theoretical work to understand and explain these behaviors.
>
> > Minor comments for improving the paper.
>
> Thank you for these detailed comments. We have addressed them in the new version of the paper.

---

### Official Review · AnonReviewer2 · 2019-10-23
**Official Blind Review #2**

**Rating:** 6

**Review:**

This paper empirically presents a very interesting connection between two also very interesting phenomena (mode connectivity and lottery ticket hypothesis), while removing a previous limitation of the lottery ticket hypothesis on larger networks. through a good amount of experiments, the authors empirically showed these two phenomena co-occur together (i.e. matching networks are stable) and have positive correlation (i.e. the more “matching” the network the more “stable”), under different network architectures and datasets.

Though it is unclear from the paper what are the immediate / straightforward applications, the findings do present interesting contributions. Several things I found that can further improve this paper.

First, this paper lacks a structured literature review. It is suggested that the findings may provide insights to our understanding of how SGD works in neural network. Laying some proper background knowledge on this area is needed in the literature review.

There are several experiments that I’m curious to see. Though I must say the existing amount of experiments sufficiently validates the existence of the connection authors put forth and hence not required.

a) Provide some metrics on how “far” are the two final weights upon which mode connectivity (or stability) is explored. For the sake of comparison, distance between the initial weights and final weights can be added.

b) First off, the introduction mentions connectivity was previously observed using “disjoint subsets” of data, whereas later in the paper only different orders of the same data are explored. I wonder if this is a typo. Regardless, exploring if the findings still apply on disjoint data and/or varying amount of data, besides different data orders, is helpful.

c) Does the full network have the property of mode connectivity (when trained using different data orders), or this only occurs under sparsity.

Lastly, the writing of the paper doesn’t interfere with understanding, but can definitely use more work. Abstract can be tightened. Several typos throughout the paper:
- in the abstract, "with the no change" -> remove "the"
- bottom of page 1, subt -> sub
- second bullet point under "contributions": remove ":"?
- page 3, paragraph starting with "stability": "the increase in worst-case increase" -> "the worst-case increase"?


**Experience Assessment:**

I have read many papers in this area.

**Review Assessment: Checking Correctness Of Derivations And Theory:**

N/A

**Review Assessment: Checking Correctness Of Experiments:**

I carefully checked the experiments.

**Review Assessment: Thoroughness In Paper Reading:**

I read the paper thoroughly.

---

> ### Author Response · Authors · 2019-11-14
> **Author Response to Reviewer 2**
>
>
> NOTE: We have posted an updated version of the paper that has been substantially restructured and rewritten to address your concerns. We highly recommend looking over the new paper.
>
> We have summarized these changes in a general response (posted as a top-level comment). We ask that you read our general response before returning to this point-by-point response. We address many of your concerns there.
>
> --------------------
>
> > It is unclear from the paper what are the immediate / straightforward applications
>
> See the PRACTICAL IMPLICATIONS section of the top-level comment.
>
> > First, this paper lacks a structured literature review.
>
> We have integrated a review of relevant literature into the body of the revised paper. For the final version of the paper, we are working on a structured literature review that we will insert after the introduction.
>
> > Does the full network have the property of mode connectivity (when trained using different data orders), or this only occurs under sparsity.
>
> Yes! In Section 3 of the new version, we show that full networks indeed have the property of mode connectivity.
>
> > Provide some metrics on how “far” are the two final weights upon which mode connectivity (or stability) is explored.
>
> In A.3 of the submission and Appendices D and E of our revised paper, we include the L2 distances between networks when we perform instability analysis. We include this data for all of our experiments (full networks and all three kinds of sparse networks). In the final version, we will also include bases for comparison (e.g., the distance between the initial and final weights, the distance between two networks trained with different initializations) to contextualize these L2 distance values.
>
> Briefly, we observe that L2 distances between the sparse networks seem to be at two different levels. When the networks are unstable (as in the case of unstable IMP subnetworks, randomly reinitialized subnetworks, and randomly pruned networks), L2 distance is at the higher level; that is, the networks are further apart. As the IMP subnetworks transition to stability, L2 distance decreases, reaching a lower (non-zero) level when they become stable. We do not observe any relationship between the stability of the unpruned networks and the L2 distances between them.
>
> > The introduction mentions connectivity was previously observed using “disjoint subsets” of data...I wonder if this is a typo.
>
> It is not a typo. The only prior work that looks at linear mode connectivity starting from the same initialization is [NK19]. That paper trains two copies of an MLP from the same initialization on disjoint subsets of MNIST. In our paper, we study different data orders rather than disjoint samples from the same distribution. We mentioned this work because wanted to give Nagarajan and Kolter ample credit since their experiment is the closest extant experiment to ours in the literature. We have emphasized this distinction in our revised paper.
>
> > Exploring if the findings still apply on disjoint data and/or varying amount of data, besides different data orders, is helpful.
>
> We agree that there are a wide range of other behaviors of neural networks that we can explore with our instability analysis framework. We are particularly interested in studying instability when training with disjoint datasets (as you mention) and when varying batch size, learning rate, network width, optimizer, and learning rate schedule (e.g., cyclic learning rates [Smith17] and exponential learning rates [LA19]). Each of these investigations could be a paper in its own right and is beyond the scope of the current work.
>
> > The writing...can definitely use more work.
>
> We have heavily revised the paper, and we believe the writing is substantially more polished. We are happy to accept further feedback that we will incorporate into the final version of the paper.
>
> [LA19] Li and Arora. An Exponential Learning Rate Schedule for Deep Learning. Arxiv.
> [NK19] Nagarajan and Kolter. Uniform convergence may be unable to explain generalization in deep learning. Arxiv.
> [Smith17] Leslie Smith. Cyclical Learning Rates for Training Neural Networks. WACV 2017.

---

### Official Review · AnonReviewer4 · 2019-11-08
**Official Blind Review #4**

**Rating:** 3

**Review:**


This paper empirically examines an interesting relationship between mode connectivity and matching sparse subnetworks (lottery ticket hypothesis).

By mode connectivity, the paper refers to a specific instance where the final trained SGD solutions are connected by a linear interpolation path without loss in test accuracy. When networks trained with SGD reliably find solutions which can be linearly interpolated without loss in test accuracy despite different data ordering,  the paper refers to these networks as ‘stable.’

Matching sparse subnetworks refer to subnetworks within a full dense network that matches the test accuracy of the full network when trained in isolation.

The paper introduces a novel improvement on the existing iterative magnitude pruning (IMP) technique that is able to find matching subnetworks even after initialization by rewinding the weights. This allowed the authors to find matching subnetworks for deeper networks and in cases where it could not be done without some intervention in learning schedule.

The paper then finds a relationship that only when the subnetworks become stable, the subnetworks become matching subnetworks.
———

Although finding a connection between two seemingly distinct phenomena is novel and interesting, I would recommend a weak reject for the following two reasons:
1) The scope of the experiment is limited to a quite specific setting,
2) there are unsupported strong claims which need to be clarified.
———

1)
In the abstract the paper claims that sparse subnetworks are matching subnetworks only when they are stable, but the results are shown in a limited setting only at a very high sparsity.
They tested stability on the highest sparsity level at which there was evidence that matching subnetworks existed, but how would the result generalize to other sparsity levels?
With lower sparsity level (if weights are pruned less), is stability easier to achieve?

The paper also focused on cases where matching subnetworks were found by IMP, but matching subnetworks can also be found by other pruning methods.
As acknowledged in the limitations section, other relationships may exist between stability and matching subnetworks found by other pruning methods, or in different sparsity levels,
which could be quite different from this paper’s claim.

In order to address this concern, I think the paper needs to show how the same relationship might generalize to different sparsity levels,
or alternatively modify the claim (to what it actually shows) and highlight the significance of the connection between matching subnetworks and stability in this highly sparse subnetwork regime.

2)
As addressed above, in the Abstract and Introduction, the paper’s claims are very general about mode connectivity and sparsity, claiming in the sparse regime, “a subnetwork is matching if and only if it is stable.” However, the experiments only show it is true in a limited setting, focusing on specific pruning method and at a specific sparsity level.
Furthermore, the statement is contradicted in Footnote 7: “for the sparsity levels we studied on VGG (low), the IMP subnetwork is stable but does not quite qualify as matching“

There are also a few other areas where there are unsupported claims.

“Namely, whenever IMP finds a matching subnetwork, test error does not increase when linearly interpolating between duplicates, meaning the subnetwork is stable.”
-> Stability was tested only at one specific sparsity level, and it is not obvious it would be stable at all lower sparsity levels where IMP found matching subnetworks.

“This result extends Nagarajan & Kolter’s observation about linear interpolation beyond MNIST to matching subnetworks found by IMP at initialization on our CIFAR10 networks”
-> Nagarajan & Kolter’s observation about linear interpolation was on a completely different setup: using same duplicate network but training on disjoint subset of data, whereas in this paper it uses different subnetworks and trains it on full dataset with different data order.

Related to the first issue, I think some of these stronger claims can be modified to describe what the experiments actually show.
The relationship found between stability and matching subnetworks in the high sparsity regime is a valuable insight that I believe should be conveyed correctly in this paper.

———

I also have some minor clarification question and suggestions for improvement.

How was the sparsity level (30%) of Resnet-50 and Inception-v3 chosen in Table 1? (which was later used in Figure 5)

— In Figure 3 and 5, the y-axis “Stability(%)” is unclear and not explained how this is computed. I first thought higher amount of stability(%) was good but it doesn't seem to be true.

— The ordering of methods for plots could be more consistent. In some figures VGG-19 come first and then Resnet-20 while for others it was the other way around, which was confusing to read. (Also same for Resnet-50 and Inception-v3)

— There are same lines in multiple graphs, but the labeling is inconsistent, potentially confusing readers:
Figure 1: (Original Init, Standard) is the same as Figure 4: (Reset),
and Figure 1: (Random Reinit, Standard) is the same as Figure 4: (Reset, Random Reinit)

**Experience Assessment:**

I have read many papers in this area.

**Review Assessment: Checking Correctness Of Derivations And Theory:**

N/A

**Review Assessment: Checking Correctness Of Experiments:**

I carefully checked the experiments.

**Review Assessment: Thoroughness In Paper Reading:**

I read the paper thoroughly.

---

> ### Author Response · Authors · 2019-11-14
> **Author Response to Reviewer 4 (Part 2)**
>
>
> > highlight the significance of the connection between matching subnetworks and stability in this highly sparse subnetwork regime.
>
> We have substantially restructured and rewritten our paper to make the significance of this connection clear. We ask that you take a look at our revised draft.
>
> > Furthermore, the statement is contradicted in Footnote 7: “for the sparsity levels we studied on VGG (low), the IMP subnetwork is stable but does not quite qualify as matching“
>
> In the submitted version of the paper, we tried to use the same sparsity level for all variants of VGG (i.e., standard, warmup, and low) and likewise for all variants of Resnet. However, our chosen sparsity level for VGG (low) was too sparse for IMP to produce a matching subnetwork at any rewinding iteration. In the updated version of the paper, we have chosen a separate sparsity level for each hyperparameter configuration based on the sparsest level for which IMP finds a matching subnetwork under any rewinding iteration we consider. We illustrate this process in Appendix A of the updated paper. The VGG (low) results now align with the other experiments.
>
> > Nagarajan & Kolter’s observation about linear interpolation was on a completely different setup: using same duplicate network but training on disjoint subset of data, whereas in this paper it uses different subnetworks and trains it on full dataset with different data order.
>
> That's correct. In the updated paper, we make sure this distinction is clear. We wanted to give Nagarajan and Kolter ample credit, since their experiment is the closest extant experiment to ours in the literature. We have emphasized this distinction in our revised paper, and we will implement any further feedback you have on clarifying this relationship to related work.
>
> > How was the sparsity level (30%) of Resnet-50 and Inception-v3 chosen in Table 1? (which was later used in Figure 5)
>
> The sparsity level is actually 70% (that is, 30% of weights remaining). These were the sparsest IMP subnetworks of Resnet-50 and Inception-v3 for which IMP found matching subnetworks at any rewinding iteration under one-shot pruning. The new Appendix A clarifies how we chose our sparsity levels for every network in the paper.
>
> > In Figure 3 and 5, the y-axis “Stability(%)” is unclear and not explained how this is computed. I first thought higher amount of stability(%) was good but it doesn't seem to be true.
>
> Calling the rise in error "stability" was a bad choice on our part. We fixed this and now call this rise in error "instability" and so lower instability is "better". Namely, when instability is 0, then the network is stable.
>
> > In some figures VGG-19 come first and then Resnet-20 while for others it was the other way around, which was confusing to read. (Also same for Resnet-50 and Inception-v3)
>
> This order is now consistent in the updated draft of the paper.
>
> > There are same lines in multiple graphs, but the labeling is inconsistent, potentially confusing readers:
>
> Labeling is now consistent in the updated draft of the paper.
>
> [FC19] Frankle and Carbin. The Lottery Ticket Hypothesis: Finding Sparse, Trainable Neural Networks. ICLR 2019.
> [HPT+15] Han et al. Learning both Weights and Connections for Efficient Neural Networks. NeurIPS 2015.
> [LSZ+19] Liu et al. Rethinking the Value of Network Pruning. ICLR 2019.
> [NK19] Nagarajan and Kolter. Uniform convergence may be unable to explain generalization in deep learning. Arxiv.

---

> ### Author Response · Authors · 2019-11-14
> **Author Response to Reviewer 4 (Part 1)**
>
>
> NOTE: We have posted an updated version of the paper that has been substantially restructured and rewritten to address your concerns. We highly recommend looking over the new paper.
>
> We have summarized these changes in a general response (posted as a top-level comment). We ask that you read our general response before returning to this point-by-point response. We address many of your concerns there.
>
> --------------------
>
> > 1) The scope of the experiment is limited to a quite specific setting,
> > The experiments only show [the relationship between mode connectivity and sparsity] is true in a limited setting, focusing on specific pruning method and at a specific sparsity level.
> > Stability was tested only at one specific sparsity level
> > The paper also focused on cases where matching subnetworks were found by IMP, but matching subnetworks can also be found by other pruning methods.
>
> We choose to focus specifically on IMP and the most extreme sparsities for which IMP can find a matching subnetworks for any rewinding iteration.
>
> Why IMP? IMP produces particularly sparse matching subnetworks and is the algorithm behind current lottery ticket results, so we are interested in studying the networks that it produces for both scientific understanding of the lottery ticket hypothesis and potential practical lessons for training extremely sparse networks to high accuracy.
>
> Why extreme sparsities? In general, sparse neural networks are more difficult to train from scratch. At extreme levels of sparsity, many classes of sparse networks (e.g., those produced by randomly reinitializing pruned networks and randomly pruning) train to lower accuracy than the full network [HPT+15, FC19, LSZ+19]. However, if it were possible to train sparse networks from scratch to the same accuracy as the full network, then it would present a new opportunity to improve the efficiency of neural network training. We are therefore interested in understanding the properties of special classes of sparse networks that are indeed matching (e.g., winning lottery tickets produced by IMP). Studying extremely sparse matching subnetworks from IMP provides the best contrast with (1) the full, overparametrized neural networks and (2) other classes of sparse networks that are not matching at these sparsities.
>
> Although we are interested in understanding this behavior at all levels of sparsity, computational limitations force us to focus on a single level of sparsity. IMP entails training a network at least a dozen times to reach high levels of sparsity, and instability analysis requires training each of these networks on three different data orders for three kinds of sparsity. For rigor, we replicate each of these experiments three times with different initializations.
>
> > 2) there are unsupported strong claims which need to be clarified.
> > In the abstract the paper claims that sparse subnetworks are matching subnetworks only when they are stable, but the results are shown in a limited setting only at a very high sparsity.
>
> As noted in the top-level comment, we have revised our claims about sparse networks to focus specifically on IMP at the most extreme sparsities for which matching subnetworks are known to exist. As we argue, IMP subnetworks at these sparsities are particularly valuable for scientific study.
>
> > They tested stability on the highest sparsity level at which there was evidence that matching subnetworks existed, but how would the result generalize to other sparsity levels? With lower sparsity level (if weights are pruned less), is stability easier to achieve?
> >  it is not obvious it would be stable at all lower sparsity levels where IMP found matching subnetworks.
>
> In short, we would not necessarily expect the results to generalize to lower sparsity levels. This is not a weakness. but just a matter of fact. As we explain in the top-level comment, the full networks are generally unstable at initialization but become stable later in training. However, they reach full accuracy regardless of whether they are stable. This means that stability and accuracy do not appear to be linked for the full network. We expect that particularly moderate sparsity levels will resemble the full network case, while higher sparsity levels will resemble the experiments in the paper.
>
> > I think the paper needs to show how the same relationship might generalize to different sparsity levels, or alternatively modify the claim (to what it actually shows)
> > Some of these stronger claims can be modified to describe what the experiments actually show.
> > The relationship found between stability and matching subnetworks in the high sparsity regime is a valuable insight that I believe should be conveyed correctly in this paper.
>
> As we discuss in the top-level comment, we have narrowed the scope of our claim to only cover  the connection between stability and matching subnetworks found by IMP in this highly sparse regime.

---

### Author Response · Authors · 2019-11-14
**Author Response - Overall Comment (Part 2)**


PRACTICAL IMPLICATIONS

Our paper is scientific in nature, with the goal of better understanding the relationship between SGD noise and the outcome of neural network optimization for both dense neural networks and sparse subnetworks found by IMP. Although our focus is not on immediate or straightforward applications, there are several ways that our results might lead to applications:

* Others have already adopted our modified version of IMP with rewinding to build practical techniques. For example, the networks generated by rewinding transfer between datasets, making it possible to train sparser networks from the start [MYP+19]. In addition, replacing fine-tuning with rewinding when pruning a neural network makes it possible to maintain full accuracy at more extreme sparsities [Anon19b]. IMP with rewinding has also been adopted to study the lottery ticket hypothesis [Anon19c, Anon19d, Anon19e, Anon19f, Anon19g].

* In larger-scale settings, we find that IMP subnetworks at extreme sparsities only become stable and matching after the full network has been trained for a small amount of time. Recent methods have explored pruning neural networks at initialization [LAT19, Anon19a], but our results suggest that the best time to prune may be slightly later in training. By that same token, most modern pruning methods only begin to sparsify networks late in training or after training [HTP+15,GEH19]. In these cases, our work suggests that there is potentially a substantial unexploited opportunity to prune neural networks much earlier in training.

* Our observations on full networks implicitly divide training into two phases: an initial, unstable phase in which the final “linearly connected” mode is undetermined on the account of SGD noise, and a subsequent, stable phase in which the final linearly connected mode becomes determined. One possible way to exploit our this observation could be to explore changing aspects of the optimization process (e.g., learning rate schedule or optimizer) once the network enters the stable phase in order to improve the performance of training. Other techniques already follow this template; for example, Goyal et al. find that warmup is necessary early in training when using large batch sizes and high learning rates [GDG17+].  Instability analysis makes it possible to evaluate the consequences of these interventions.

SUMMARY OF TECHNICAL CHANGES

* We have renamed “stability” to “instability” so that a network is “stable” when “instability” is 0.

* We have moved results on full networks from the appendices into the main body of the paper as Section 3.

* In our analysis of full networks, we have examined instability with respect to train error in addition to test error.

* We have updated our implementations of Resnet-20 and VGG-16 to reach higher, state-of-the-art accuracy. At this higher accuracy, the IMP subnetworks of these networks now become stable and matching slightly later in training than before, but they still do so 1-2% into training.

* We compare the instability of IMP subnetworks to that of randomly reinitialized IMP subnetworks in addition to randomly pruned subnetworks.

[Anon19a] Anonymous. Picking Winning Tickets Before Training by Preserving Gradient Flow. In submission to ICLR 2020.
[Anon19b] Anonymous. Comparing Fine-Tuning and Rewinding in Neural Network Pruning. In submission to ICLR 2020.
[Anon19c] Anonymous. Playing the Lottery with Rewards and Multiple Languages. In submission to ICLR 2020.
[Anon19d] Anonymous. Finding Winning Tickets with Limited (or No) Supervision. In submission to ICLR 2020.
[Anon19e] Anonymous. Winning the Lottery with Continuous Sparsification. In submission to ICLR 2020.
[Anon19f] Anonymous. The Sooner the Better: Investigating the Structure of Early Winning Lottery Tickets. In submission to ICLR 2020.
[Anon19g] Anonymous. The Early Phase of Neural Network Training. In submission to ICLR 2020.
[GDG+17] Goyal et al. Accurate, Large Minbatch SGD: Training Imagenet in 1 Hour. CVPR 2018.
[GEH19] Gale et al. The State of Sparsity in Deep Neural Networks. Arxiv.
[HPT+15] Han et al. Learning both Weights and Connections for Efficient Neural Networks. NeurIPS 2015.
[LAT19] Lee et al. SNIP: Single-Shot Network Pruning Based on Connection Sensitivity. ICLR 2019.
[MYP+19] Morcos et al. One Ticket to Win them All: Generalizing Lottery Ticket Initializations Across Datasets and Optimizers. NeuIPS 2019.

---

### Author Response · Authors · 2019-11-14
**Author Response - Overall Comment (Part 1)**


NOTE: We have posted an updated version of the paper that has been substantially restructured and rewritten to address your concerns. We highly recommend looking over the new paper.

--------------------

We thank the reviewers for their feedback.

Upon reading reviews 2 and 3, we recognize that we failed to adequately communicate the significance of our results. Upon reading review 4, we recognize that we failed to clarify the scope of our claims and justify the importance of our chosen methodology.

Based on your feedback, we have substantially restructured and rewritten our paper to address these concerns. We believe that our “stability analysis” framework (which as per R4 we now call “instability analysis”) and our new observations about IMP subnetworks are significant contributions, and we hope to convince you that this is the case with our updated version.

Here, we summarize our revised framing. We have responded to specific concerns of individual reviewers in separate replies to their comments.

SUMMARY OF REVISED FRAMING

In our original submission, we framed our contribution as a surprising connection between two empirical phenomena of recent interest: mode connectivity and sparse neural networks in the context of the lottery ticket hypothesis.

In the revised version, we instead emphasize that our “instability analysis” framework provides a new lens through which to study the behavior of neural networks by way of linear mode connectivity.

We demonstrate the value of this framework in two ways. First, we study the instability of full, unpruned networks. We now recognize that this data, which was previously buried in the appendices, is a significant contribution in its own right and an important part of our story. The central finding of this experiment is that all networks become stable early in training. That is, early in training, the outcome of optimization is determined modulo linear mode connectivity.

We then use instability analysis to better understand “lottery ticket” networks found by IMP. Our core finding is that, at extreme sparsities, an IMP subnetwork is matching (i.e., it can train in isolation to full accuracy) only when it is stable. This insight provides the first basis for understanding the mixed results on IMP in the literature. In addition, we modify IMP to “rewind” subnetworks to their values at an iteration k > 0 rather than to initialization. For values of k that are early in training, IMP subnetworks become stable and matching in all cases that we consider, including large-scale settings where IMP fails to do so at initialization.  In response to R4’s suggestions, we have modified the scope of our claims to focus exclusively on IMP subnetworks at the highest sparsity for which IMP at any rewinding iteration produces a matching subnetwork.

CONTRIBUTIONS AND IMPLICATIONS

Our revisions aim to clarify that our work makes significant contributions to both (1) our understanding of SGD on neural networks and (2) our understanding of sparse IMP subnetworks and the lottery ticket hypothesis. Please see the updated “Contributions” paragraph in our new introduction.

---

### Decision · Program_Chairs · 2019-12-19

**Decision:**

Reject

**Comment:**

This paper investigates theories related to networks sparsification, related to mode connectivity and the so-called lottery ticket hypothesis.  The paper is interesting and has merit, but on balance I find the contributions not sufficiently clear to warrant acceptance.  The authors made substantial changes to the paper which are admirable and which bring it to borderline status.